# Epiregulin as an Alternative Ligand for Leptin Receptor Alleviates Glucose Intolerance without Change in Obesity

**DOI:** 10.3390/cells11030425

**Published:** 2022-01-26

**Authors:** No-Joon Song, Aejin Lee, Rumana Yasmeen, Qiwen Shen, Kefeng Yang, Shashi Bhushan Kumar, Danah Muhanna, Shanvanth Arnipalli, Sabrena F. Noria, Bradley J. Needleman, Jeffrey W. Hazey, Dean J. Mikami, Joana Ortega-Anaya, Rafael Jiménez-Flores, Jeremy Prokop, Ouliana Ziouzenkova

**Affiliations:** 1Department of Human Sciences, The Ohio State University, Columbus, OH 43210, USA; Nojoon.Song@osumc.edu (N.-J.S.); lee.7278@buckeyemail.osu.edu (A.L.); yasmeen.2@buckeyemail.osu.edu (R.Y.); shenshenqiwen@gmail.com (Q.S.); yangkf@sjtu.edu.cn (K.Y.); drshashikumar81@gmail.com (S.B.K.); muhanna.2@buckeyemail.osu.edu (D.M.); arnipalli.1@buckeyemail.osu.edu (S.A.); 2Department of Nutrition, School of Medical, Shanghai Jiao Tong University, Shanghai 200025, China; 3Division of General and Gastrointestinal Surgery, Center for Minimally Invasive Surgery, The Ohio State University, Columbus, OH 43210, USA; Sabrena.Noria@osumc.edu (S.F.N.); needleman.3@osu.edu (B.J.N.); 4Department of Surgery, The Ohio State University, Columbus, OH 43210, USA; Jeffrey.hazey@memorialohio.com; 5Department of Surgery, University of Hawaii, Honolulu, HI 96813, USA; dmikami2@hawaii.edu; 6Department of Food Science and Technology, The Ohio State University, Columbus, OH 43210, USA; joana_oa@outlook.com (J.O.-A.); jimenez-flores.1@osu.edu (R.J.-F.); 7Department of Pediatrics and Human Development, College of Human Medicine, Michigan State University, Grand Rapids, MI 49503, USA; jprokop54@gmail.com

**Keywords:** epiregulin, leptin receptor, ERK, EGFR, glucose uptake, energy metabolism

## Abstract

The leptin receptor (LepR) acts as a signaling nexus for the regulation of glucose uptake and obesity, among other metabolic responses. The functional role of LepR under leptin-deficient conditions remains unclear. This study reports that epiregulin (EREG) governed glucose uptake in vitro and in vivo in *Lep^ob^* mice by activating LepR under leptin-deficient conditions. Single and long-term treatment with EREG effectively rescued glucose intolerance in comparative insulin and EREG tolerance tests in *Lep^ob^* mice. The immunoprecipitation study revealed binding between EREG and LepR in adipose tissue of *Lep^ob^* mice. EREG/LepR regulated glucose uptake without changes in obesity in *Lep^ob^* mice via mechanisms, including ERK activation and translocation of GLUT4 to the cell surface. EREG-dependent glucose uptake was abolished in *Lepr^db^* mice which supports a key role of LepR in this process. In contrast, inhibition of the canonical epidermal growth factor receptor (EGFR) pathway implicated in other EREG responses, increased glucose uptake. Our data provide a basis for understanding glycemic responses of EREG that are dependent on LepR unlike functions mediated by EGFR, including leptin secretion, thermogenesis, pain, growth, and other responses. The computational analysis identified a conserved amino acid sequence, supporting an evolutionary role of EREG as an alternative LepR ligand.

## 1. Introduction

Leptin receptor (LepR, Alias: CD295, or ObR) regulates critical aspects of energy homeostasis, appetite, and notably plays a central role in the regulation of glucose uptake in both peripheral organs and the central nervous system [1,2]. LepR activation is mediated by the adipokine leptin, which is the only currently known ligand for LepR [2]. The malfunction of leptin and/or LepR is among the most widespread metabolic malfunctions associated with genetic, immune, endocrine, and diet-related diseases. Deficiencies in leptin or LepR signaling, or leptin resistance can develop in response to inflammation, endocrine disorders, lipodystrophy, or obesity, which progresses to glucose intolerance and insulin resistance [2,3]. In animal models, a genetic LepR deficiency in *Lepr^db^* mice manifests with obesity, diabetes, hyperglycemia, insulin resistance, as well as aberrant immune responses and reproduction [2]. Leptin deficiency in *Lep^ob^* mice presents a less severe phenotype than LepR-deficient *Lepr^db^* mice [4]. These differences indicate a possible existence of other factors interacting with LepR via direct or indirect mechanisms.

Leptin binding drives conformational activation of a LepR multimeric complex and exhibits high evolutionary conservation within the Unpaired (UPD)/Domeless (DOME) cytokine-receptor interaction in drosophila [5]. In mammals, leptin binds to cytokine receptor homology domain 2 (CHR2), which interacts with an immunoglobulin-like domain (IGD) [6]. Multimeric complex formation of the intracellular domain of LepR [7] is critical in activating Janus-kinase 2 (JAK2) and initiating downstream signaling via signal transducer and activator of transcription 3 (STAT3) and 5 (STAT5) [8,9], phosphoinositide 3-kinase (PI3K), mitogen-activated protein kinase (MAPK), and mTOR kinases. The length of the intracellular domain of LepR isoforms determines the variety of signaling and pleiotropic biological activity. Long cytosolic domains act predominantly in the brain, where they regulate appetite, energy homeostasis, and crosstalk with the insulin pathway [6]. LepR with short intracellular domains is expressed in peripheral tissues [6], where they regulate glucose uptake in response to leptin or pharmacological ligands [10]. Different roles of leptin and LepR in the context of glucose uptake were highlighted in studies using *Lep^ob^* [11] and *LepR^db^* mice [12]. Caloric restriction normalized weight in both leptin and LepR deficient mice; however, hyperglycemia was only improved in *Lep^ob^* [11] but not in *LepR^db^* mice [12]. These differences in glucose uptake under leptin and LepR deficiencies suggest the presence of other activator(s) of LepR that could stimulate glucose uptake in the absence of leptin.

The LepR response can also be the result of cross-activation of multiprotein complexes with other extracellular receptors including epidermal growth factor (EGF) receptor (EGFR, alias: ERBB1, HER1), insulin-like growth factor I receptor (IGF-IR), lipoprotein receptor-related protein 1 and 2 (LRP1 and LRP2), and vascular endothelial growth factor receptor (VEGFR) [13]. Particularly, EGFR can form a heterodimeric complex with LepR via immunoglobulin-like domain (IGD) or fibronectin-like (FN) III domain without ligand participation [6], can play a role in physiological growth and development, and determine the risk for carcinogenesis [14,15]. The cross-activation can also depend on several ligands for EGFR, including EGF and epiregulin (EREG) [16]. EREG was purified from a secretome of mouse NIH3T3 fibroblasts as a 46-amino-acid single chin (Val^1^-Leu^46^) polypeptide by Toyoda and colleagues [17] and characterized as a low-affinity ligand for EGFR with growth-promoting properties in fibroblasts and growth-suppressing properties in epithelial tumor cells. Three disulfide bridges stabilize the EREG structure comprising of an N-terminal domain (Ile^3^–Glu^33^), a core region with β-hairpin motif (Gly^17^–Cys^32^), and a C-terminal domain (Val^34^–Phe^45^) [18]. In the context of EGFR activation, the role of EREG was studied in many cancers and remains controversial (reviewed in [19]), because either protective [20,21] or pathogenic effects [22] were demonstrated. Systematic investigation of *Ereg* expression levels in humans under physiological conditions was reported in Human Protein Atlas [23]. These investigations revealed the predominant expression of *Ereg* in kidney fibroblasts as well as in neutrophils and resident macrophages in many tissues with the highest levels of *Ereg* expression in tissues of mesenchymal origin such as bone marrow, lymphoid tissue, and skin. In our previous work, the EREG/EGFR signaling cascade was shown to regulate leptin secretion and activation of LepR to stimulate thermogenesis in white adipose tissue and weight loss [24]. Thermogenic adipocytes have an increased glucose demand that was associated with the increased secretion of EREG, but not the other EGFR ligands [25] (GEO file: ‘QS wild type and *Aldh1a1* KO preadipocytes 2015′). EREG secretion is also a hallmark of other cells with high glucose demands including stem cells, various cancers, and inflammatory cells [26]. However, the crosstalk of EREG with LepR and/or EGFR and its implication for glucose metabolism has not been systematically investigated. The findings in this study revealed an interaction between EREG and LepR that constitutes a novel and efficient glucose uptake pathway.

## 2. Materials and Methods

### 2.1. Reagents

All reagents were purchased from Sigma-Aldrich and all cell culture media were purchased from Life Technologies/ThermoFisher Scientific (Waltham, MA, USA) unless otherwise indicated. Mouse recombinant EREG (50599-M01H, Sino Biological Beijing, China) or Creative Biomart (No. Ereg-576M, New York, NY, USA) and human recombinant EREG (1195-EP/CF, R&D Systems, Minneapolis, MN, USA) were used for in vitro assays and/or in vivo studies. Mouse recombinant EGF was purchased from ThermoFisher Scientific (Cat. No. PMG8041, Waltham, MA, USA) and mouse recombinant leptin was purchased from PeproTech (Cat. No.450-31 Rocky Hill, NJ, USA). Human recombinant insulin was obtained from Sigma-Aldrich and human recombinant leptin was purchased from Invitrogen. We used EGFR inhibitor (Tyrphostin AG 1478, Cat. No.4774, sc-200613, Santa Cruz Biotechnology, CAS 175178-82-2, Sigma-Aldrich, Saint Louis, MO, USA), irreversible inhibitor of EGFR and ErbB2 (Allitinib (AST-1306, CAS 1050500-29-2), Cat. No. S2185, SelleckChem, Houston, TX, USA), pan-ErbB inhibitor for EGFR and ErbB2 (Canertinib, CI-1033, Cat. No. S1019 (CAS 267243-28-7); SelleckChem), dual IR/IGF-1R inhibitor (BMS 536924 (CAS 468740-43-4), Cat. No.4774, Tocris Bioscience, Bristol, UK,), and SRC inhibitor (AZM475271 Cat. No.39-631-0 (CAS476159-98-5), ThermoFisher Scientific). ERK inhibitors were the non-competitive inhibitors of MEK-1 and MEK-2 (U0126 Cat. No.1144 (CAS 109511-58-2), Tocris Bioscience), specific inhibitor of ERK1/2 (SCH772984 (CAS 942183-80-4), Cat. No. S7101, Selleck Chemicals, Houston, TX, USA), and water-soluble ERK dimerization inhibitor (DEL-22379 (CAS181223-80-3), Cat. No. S7921, Selleck Chemicals). PI3K (Wortmannin) inhibitor was purchased from Cayman Chemical (Cat. No. 10010591, CAS 19545-26-7 Ann Arbor, MI, USA). MAPK (U0126, CAS 109511-58-2), inhibitors were from Tocris Bioscience (Cat. No. 1144).

### 2.2. Human Tissues

This study was approved by the Ohio State University Institutional Review Board (IRB). All subjects provided written informed consent for tissue and data collection. Intra-abdominal (iAb) fat was obtained from the greater omentum of the peritoneum during bariatric surgeries (laparoscopic banding and gastric bypass) in obese patients (BMI = 50 ± 8.4, *n* = 7 overnight fasted men and women, aged 40 ± 11 years) with or without type 2 diabetes mellitus (HbA1C 6 ± 1). The exclusion criteria included: (1) Previous gastric bypass surgery; (2) pregnancy; (3) recent malignancy (within 6 months) or any history of chest radiation; and (4) recent (3 months) history of steroid or immunosuppressive agent use. 

### 2.3. Animal Studies

Animal studies were approved by the Institutional Animal Care and Use Committee of The Ohio State University (OSU). 

Leptin-deficient mouse model. Six-week-old *Lep^ob^* male mice (B6.V-Lepob/J, the Jackson Laboratory (stock number 000632, *n* = 14)). *Lep^ob^* mice were fed a regular chow diet (Teklad LM-485 mouse/rat diet, irradiated; Envigo, Somerset, NJ, USA) throughout the study (26 days). Mice were randomly assigned into two groups: (1)Control *Lep^ob^* mice group, injected with 0.1 mL sterile PBS (*n* = 7), and(2)EREG-treated group of *Lep^ob^* mice (*n* = 7), injected intraperitoneally with PBS containing EREG (50 ng/g body weight (BW)). EREG was injected every other day for 26 days. Echo-MRI analysis was performed at the beginning and the end of the study. Blood was collected by cardiac puncture at the end of the study.

Leptin receptor-deficient mouse model. Five-week-old *Lepr^db^* male mice (BKS.Cg-*Dock7^m^* +/+ *Lepr^db^*/J; homozygous for *Lepr^db^*) were purchased from Jackson Laboratory (stock number 000642, *n* = 12). Mice were randomly assigned to a control group treated with PBS (Veh; 10 µL/g body weight, *n* = 6) or EREG treatment group (50 ng/g body weight, *n* = 6) by intrascapular injections every other day for 4 weeks. 

### 2.4. Glucose Tolerance Test (GTT) 

Chronic exposure study. GTT was performed in fasted *Lep^ob^* (*n* = 7 per group) or *Lepr^db^* (*n* = 6 per group) mice at the end of the study. After intraperitoneal injection of 1 mg glucose/g BW, blood glucose levels were measured at indicated time points. The area under the curve (AUC) was calculated using GraphPad Prism 7 (GraphPad Software, Inc., San Diego, CA, USA). Blood glucose levels were measured from mouse tails by One Touch Ultra glucometer (LifeScan, Chesterbrook, PA, USA).

Single exposure GTT test. Another group of five-week-old *Lep^ob^* male mice was used to analyze the effect of a single dose of EREG. GTT was also performed in fasted *Lep^ob^* injected i.p. with PBS (Veh; 10 µL/g body weight, *n* = 5) or EREG-treatment group (50 ng/g body weight, *n* = 5) in addition to the injection with 1 mg glucose/g BW. 

### 2.5. EREG and Insulin Tolerance Tests (Single Exposure)

Fifteen non-treated *Lep^ob^* male mice were randomized into three groups, Veh-, insulin-, and EREG-treated (*n* = 5/group). Prior to testing, mice were fasted for 4 h. For insulin and EREG tolerance tests, we used intraperitoneal injection of 0.012 IU insulin/g BW and 80 ng EREG/g BW, respectively. Blood glucose level was measured at indicated time points.

### 2.6. Body Composition Analysis

Body composition was measured in living mice using the Body Composition Analyzer for Live Small Animals (EchoMRI™-100H, Houston, TX, USA) as described [10].

### 2.7. Adipocyte Culture

Murine preadipocytes (3T3-L1 fibroblasts, CL-173 from American Type Culture Collection) were cultured as described [10]. Human stromal vascular fraction (SVF) was isolated from the visceral (omental) fat of obese subjects using type 1 collagenase (17100017, Thermo Fisher Scientific) and cultured as described [27]. 

### 2.8. Glucose Uptake Assay

Glucose uptake assay was performed using glucose uptake cell-based assay kit (Cayman Chemical, 600470) described in [10]. 2-deoxy-2-[(7-nitro-2,1,3-benzoxadiazol-4-yl)amino]-D-glucose (FD-Glucose) containing medium with or without inhibitors was added to the plated cells in glucose-free DMEM. Fluorescence at an excitation/emission of 485/535 nm was measured using the Synergy H1 Hybrid Multi-Mode Microplate Reader (BioTek, Winooski, VT, USA).

### 2.9. Transient Transfection and GLUT4 Translocation Measurement

pB-GLUT4-7myc-GFP was a gift from Jonathan Bogan (Addgene, Cambridge, MA, USA, plasmid # 52872). Transient transfection of NIH-3T3 cells, was performed as previously described [28]. Plasmid DNA (0.9 μg/chamber) and FuGENE 6 Transfection Reagent (Promega, Madison, WI, USA) (1:3.5 ratio, *v*/*v*) were incubated in Opti-MEMI medium (Invitrogen, Grand Island, NY, USA) for 45 min at room temperature. Cells transfected for 16 h were starved for 40 min in glucose-free DMEM medium and stimulated with PBS (vehicle), insulin (10 µg/mL), or EREG (50 ng/mL) for 60 min. Fluorescence was measured in formalin-fixed cells using the EVOS FL Cell Imaging System (Life Technologies, Waltham, MA, USA) at 10× magnification. Signal intensity was analyzed by ImageJ software (imagej.nih.gov/ij/download/, accessed on 1 January 2022)

### 2.10. Western Blot 

Protein lysates were separated on 4–20% gradient SDS-PAGE (Bio-Rad, Hercules, CA, USA) and transferred to a PVDF membrane (Bio-Rad). Antibodies against mouse AKT (4691S), phosphorylated AKT (p-AKT, 9271S), STAT5 (94205S), p-STAT5 (4322S), STAT3 (9139S), ERK (4696S), and p-ERK (4370S) were purchased from Cell Signaling Technology (Danvers, MA). The antibody against LepR and β-actin were acquired from ThermoFisher Scientific (PA1-28844, LOT# SH2429627A) and Sigma-Aldrich (A5441). Detection was performed using an Odyssey Infrared Imaging System (LI-COR Biosciences). Images were quantified by ImageJ software.

### 2.11. Immunoprecipitation

Adipose tissue from male *Lep^ob^* male mice (400 mg) was homogenized in RIPA buffer (Thermo Scientific, Waltham, MA, USA) and incubated at 4 °C for 1 h. Every 15 min, protein lysates were gently vortexed and after 1 h incubation, samples were centrifuged to collect supernatants. Antibodies specific to EREG or LepR were conjugated with dynabeads (14321D, Thermo Fisher Scientific) for 16 h according to the manufacturer’s protocol. Antibodies conjugated with dynabeads were incubated with protein lysates for 4 h to maximize the protein-antibody binding. Mouse EREG antibody (sc-376284) was purchased from Santa Cruz Biotechnology (Dallas, TX, USA) and mouse LepR antibody (20966-1-AP) was purchased from (ProteinTech, Rosemont, IL, USA). IgG control was also conjugated to dynabeads and incubated with protein lysates as a co-IP control. Protein complexes were further purified by magnets and eluted by elution buffer. Eluted proteins were further analyzed by Western blot to confirm the direct binding of proteins. After blocking, EREG and EGF were treated for 30 min. Polyclonal antibodies (Catalog # PA5-18522) recognizing human and mouse LepR were purchased from Invitrogen/ThermoFisher Scientific.

### 2.12. Enzyme-Linked Immunosorbent Assay (ELISA)

Mouse insulin and leptin levels were measured with an ELISA kit purchased from EMD Millipore (EZRMI-13K) and Crystal Chem (Elk Grove Village, IL, USA, 90030), respectively.

### 2.13. Epiregulin Protein Modeling 

Evolutionary analysis of EREG was performed using our previously published pipeline [29]. Epiregulin interaction with EGFR was shown using PDB structure 5wb7. Docking of Epiregulin to LepR was performed using HADDOCK2.2 [30] and our previous model of Lep-LepR 2:2 [5].

### 2.14. Quartz Crystal Microbalance with Dissipation (QCM-D) Binding Assay 

QCM-D binding assay was performed as described previously [10]. Briefly, a quartz sensor with a frequency of 4.95 MHz of an active gold surface (QSX 301, Biolin Scientific, Gothenburg, Sweden) was equilibrated with PBS buffer (flow rate, 50 µL/min, pH 7.1–7.5, 25 °C) using a Q-Sense Explorer (Biolin Scientific). Mouse recombinant leptin receptor (LepR) protein (1.6 pM in PBS; R&D systems, 497-LR/CF) was added to establish a monolayer, followed by the addition of mouse leptin (1.6 fM in PBS; Crystal Chem, Elk Grove Village, IL, USA, 90030-B) or mouse EREG (1.6 fM). The interaction of LepR with leptin or EREG was measured as the difference in frequency (∆F) and dissipation (∆D) values of the odd overtones, and the thickness of the film was modeled using Voight–Voinova equations for homogenous viscoelastic layers [31] for homogenous viscoelastic layers assuming a fixed density of 1 g/mL using QTools 3 software (Biolin Scientific).

### 2.15. Statistical Analysis

All data were analyzed using SPSS 23 (IBM Corp. in Armonk, NY, USA). All data are shown as the mean ± SEM. The number of samples is indicated in figure legends. Group comparisons were assessed using ANOVA models or Student’s *t*-test, unless otherwise noted. *p* < 0.05 was considered statistically significant.

## 3. Results

### 3.1. EREG Regulated Glucose Metabolism under Leptin Deficient Conditions but Required LepR 

We investigated the glycemic effects of EREG in *Lep^ob^* mice, which have functional LepR but lack leptin. Randomized *Lep^ob^* male mice were treated with vehicle (PBS) or EREG (50 ng/g body weight) for 4 weeks and had ad libitum access to the regular chow diet. Both control and EREG-treated mice gained similar weight (Figure 1A) and had similar food intake (Appendix A). EREG-treated and control groups also had similar body fat compositions (Figure 1B). The proportion of lean mass was modestly increased by EREG treatment (43% vs. 40% in control) (Figure 1C). These data demonstrate that leptin was required for energy balance and EREG did not modulate this process. 

To assess glucose utilization in these mice, we performed a glucose tolerance test (GTT, Figure 1D,E). In contrast to the minor changes in body weight and composition, glucose intolerance was markedly improved in *Lep^ob^* mice treated with EREG (60% vs. 100% in control, Figure 1E). The changes in plasma levels of insulin were not significant (Figure 1F), although the insulinotropic activity of EREG was found in vitro [32]. A similar decrease in glucose uptake (Appendix A) without changes in plasma insulin levels (Appendix A) was also observed in our previous study in *Lep^ob^* males fed a high-fat diet [24]. A high-fat diet also did not influence weight changes in EREG-treated and control *Lep^ob^* males [24]. The glycemic effects of EREG were distinct from the other effects of EREG, including body weight, metabolic rate, and thermogenesis, which are dependent on EGFR receptor and leptin secretion [24]. Thus, EREG improved glucose uptake under leptin-deficient conditions without affecting other leptin-dependent metabolic pathways such as satiety and energy balance.

The dependence of glycemic effects of EREG on LepR was examined in LepR-deficient *Lepr^db^* male mice treated with or without EREG (50 ng/g body weight) for 35 days. Both groups of mice gained weight significantly; however, EREG-treated mice had moderately increased weight (108%) compared to the control group (100%) (Figure 1G). Moreover, the composition of fat (Figure 1H) and lean mass (Figure 1I) was similar in these groups. EREG treatment also did not alter glucose uptake, according to the results of GTT testing (Figure 1J). Additionally, fasting glucose kinetics were not influenced by EREG treatment (Appendix A). *Lepr^db^* mice in both groups had similar plasma levels of insulin (Figure 1J,K) and leptin (Appendix A). Therefore, LepR deficiency abolished the glycemic action of EREG, suggesting the involvement of LepR in EREG-induced glucose uptake.

### 3.2. EREG Binds to LepR

Next, we compared the effects of a single dose of EREG (50 ng/mL) and insulin (12 IU/kg) using respective tolerance tests in two groups of *Lep^ob^* mice (*n* = 5/group) without any prior treatment (Figure 2A). The changes in blood glucose concentrations in response to insulin injection were not significant compared to the initial levels in the course of the insulin tolerance test (ITT). This observation was expected, given reported insulin resistance in *Lep^ob^* mice [2]. Transiently, 30 min post-injection, a greater decrease in the blood glucose was seen in *Lep^ob^* mice undergoing ITT compared to EREG tolerance tests. This trend was reversed at 60 and 90 min after EREG injections and resulted in a significant and robust decrease in glucose concentrations compared to the initial levels in *Lep^ob^* mice, although the area under the curve (AUC) was not different from the insulin-induced change in glucose uptake (Figure 2B). These experiments demonstrate the sensitivity of EREG in the late phase regulation of glucose in *Lep^ob^* mice under leptin-deficient conditions.

We further tested the effects of a single EREG dose on glucose metabolism using the GTT test. *Lep^ob^* mice were injected i.p. with or without EREG (50 ng/g body weight) in addition to glucose (Figure 2C). In the GTT test, EREG-treated mice exhibited significantly reduced glucose levels in blood 30 min after treatment compared to the control group of *Lep^ob^* mice. GTT quantification using AUC revealed significantly reduced glucose levels in response to EREG treatment vs. control group (Figure 2D). Given the ability of EREG to directly induce pathways responsible for glucose uptake in the absence of leptin but not LepR, next, we examined the binding between EREG and LepR using immunoprecipitation.

We elucidated the binding between endogenous EREG and LepR in *Lep^ob^* mice under non-stimulated and stimulated conditions (Figure 2E,F). We immunoprecipitated the EREG complex with its endogenous binding partners from subcutaneous (Figure 2E) and visceral (Figure 2F) fat pads 15 min after EREG injection and compared them to those in non-treated *Lep^ob^* male mice. Western blot analysis revealed that the EREG complex contained endogenous LepR in both subcutaneous and visceral fat. In subcutaneous adipose tissues from EREG-stimulated mice, we detected 80% higher levels of EREG complex with LepR compared to the levels found in non-treated *Lep^ob^* mice; however, in visceral fat, expressing low levels of LepR [33], this effect was diminished. The EREG co-precipitation was validated in an obese insulin-resistant patient representing a cohort with well-documented low levels of LEPR expression [34]. Immunoprecipitation was performed in abdominal omental adipose tissue (Appendix A). The immunoprecipitated complex with human anti-EREG antibody showed very low expression levels of LEPR. Together, these data suggest that the glycemic function of EREG was associated with the binding of EREG to LepR under leptin-deficient conditions, and that it was augmented after treatment with EREG.

### 3.3. EREG Stimulation of Glucose Uptake In Vitro Depends on LepR

Regulation of systemic energy homeostasis is attributed to hypothalamic activation of the long LepR by leptin [1,35,36]. Physiological expression of *Ereg* was reported in fibroblasts isolated from adipose tissue, bone marrow, and other tissues [37]. We next examined the effects of EREG on glucose uptake in primary cultures of human stromal vascular fraction (SVF) cells, isolated from the omental fat of seven different subjects (Appendix A). Glucose uptake was measured using the widely established fluorescent derivative of glucose (FD-glucose) [38]. In these human SVF cells, EREG stimulation was effective and significantly induced FD-glucose uptake. EREG reached similar levels as the FD-glucose uptake stimulated by insulin or leptin; however, this effect was achieved at lower concentrations compared to insulin or leptin (EREG 25–100 ng/mL vs. insulin 10 µg/mL or leptin 200 ng/mL). The underlying role of peripheral LepR in glycemic effects of EREG was further examined in SVF cells isolated from *Lepr^db^* mice (Figure 3A). Leptin and insulin lost their ability to stimulate glucose uptake in the absence of LepR in these cells, which was in agreement with previous studies [13]. EREG-mediated glucose uptake was also inhibited, suggesting that EREG responses were dependent on LepR.

Next, we examined FD-glucose uptake in SVF cells from adipose tissue isolated from *Lep^ob^* mice in the presence and absence of the EGFR inhibitor (Figure 3B). In these cells, stimulation with EREG significantly increased glucose uptake (Figure 3B). Both leptin and insulin stimulated glucose uptake in these cells, as expected. EREG is one of the established ligands for EGFR, which regulates leptin secretion, thermogenesis, growth, and other effects in response to EREG stimulation [20,24,39]. Surprisingly, inhibition of EGFR (EGFR-I) significantly increased glucose uptake by EREG in *Lep^ob^* SVF cells. Cumulatively, glycemic effects of EREG were not impaired by EGFR inhibition; however, they required LepR, which was in agreement with our observations in vivo.

3T3-L1 preadipocytes are an established cell culture model for adipogenesis and glucose metabolism [40]. Similar to primary SVF cells, 3T3-L1 preadipocytes increased uptake of FD-glucose in response to EREG or insulin stimulation (Figure 3C). EREG regulated glucose uptake in a time-dependent (Figure 3D) and dose-dependent manner (EC_50_ = 26 nM, Figure 3E) in these cells. The major glucose transporter (GLUT) in adipose tissue and 3T3-L1 cells is the isotype GLUT4 [41]. Both EREG and insulin increased translocation of the fluorescently labeled GLUT4 transporter to the plasma membrane for glucose transport (Figure 3F), concurrent with their efficacy for glucose uptake stimulation (Figure 3G). Other pathways, such as IGF1/lipoprotein receptor-related protein 1 (LRP1) control GLUT1-dependent glucose uptake in the nervous tissue [42]. However, the deficiency in LRP1 did not influence FD-glucose uptake in the WAT in mice with LRP1-inactivated WAT [43], suggesting a minor role of this pathway in WAT. Nonetheless, mechanisms underlying EREG-dependent regulation of glucose uptake are likely different between WAT and other tissues, but their investigation was beyond the scope of this study.

To investigate systematically the involvement of additional relevant receptors and kinases in EREG-mediated glucose uptake in 3T3-L1 cells we employed various inhibitors (Figure 4A,B). The family of ERB receptors, including EGFR and Her2, was inhibited by Canertinib CL1033, Tyrphostin, and AST1306, respectively. Canertinib is a pan-ErbB inhibitor for EGFR (IC50 1.5 nM) and ErbB2 (IC50 9.0 nM), which had no activity with PDGFR, FGFR, InsR, PKC, or CDK1/2/4 [44]. Tyrphostin is a selective EGFR tyrosine kinase inhibitor (IC50 = 3 nM) [45]; whereas AST1306 (IC50 = 12 nM) inhibited EGFR and ErbB2 by irreversible covalent binding with Cys797 and Cys805 in the catalytic domains without exhibiting activity on Akt, Src, Jak2, and other kinases [46]. These inhibitors did not block glucose uptake mediated by EREG treatment (Figure 4A). The EREG also did not stimulate glucose uptake in the presence of inhibitors of insulin/IGF receptors (BMS 536924) and Src tyrosine kinase (AZM475271) (Figure 4A). BMS 536924 inhibits both insulin-like growth factor receptor (IGF-1R) kinase (IC50 = 100 nM) and InsR (IC50 = 73 nM) [47]. AZM475271 is a potent and selective Src kinase inhibitor (IC50 = 5 nM) with no inhibitory activity on fms-like tyrosine kinase 3 (Flt3), VEGF, and angiopoietin Tie-2 receptor [48].In line with previously reported findings [49,50], chemical inhibition of EGFR or SRC increased glucose uptake, but this uptake was not further stimulated by EREG. Notably, the increase in glucose uptake by EGF, which was previously attributed to the activation of EGFR [51], was completely inhibited by antibodies against EREG (Appendix A). Endogenous EREG may be a confounding factor contributing to the glycemic effects associated with the EGF/EGFR pathway [50,51]. EGFR inhibition streamlines activation of EREG-dependent pathways for glucose uptake that results in the improved glucose uptake in the presence of EGFR inhibitors. Collectively, these studies suggest that glycemic effects of EREG are dependent on LepR and not on EGFR. 

### 3.4. EREG Stimulated Glucose Uptake via the ERK/PIK3 Signaling Pathway

We examined the role of EREG in the activation of downstream signaling pathways associated with EGFR and LepR in 3T3-L1 cells (Figure 4B). MEK1/2 was ruled out as a mediator for the effects of EREG on glucose uptake, based on the experiments with MEK1/2 inhibitors (Figure 4B). In contrast, inhibition of the PI3K pathway abolished glucose uptake mediated by EREG, consistent with the central role of this kinase in pathways regulating glucose metabolism [52] (Figure 4B). EREG responses in primary human SVF cells from different donors were consistent with those observed in 3T3-L1 cells. In SVF cells, inhibition of PI3K blocked EREG-dependent glucose uptake with or without MEK1/2 inhibitors (Appendix A); however, inhibition of EGFR or MEK1/2 alone promoted glucose uptake in the presence of EREG (Appendix A). 

PI3K acts in conjunction with AKT. However, dose-dependent EREG stimulation was not associated with early AKT phosphorylation in 3T3-L1 cells (Figure 4C and Appendix A). AKT phosphorylation levels were similar after 5 and 15 min stimulation with different EREG. AKT phosphorylation is also required for Rab GTPase activating protein (Alisa: Akt substrate of 160 kDa, AS160, or TBC1D4) [53]. Although AS160 increases the protein content of GLUT4 in WAT, AS160 deficiency did not prevent glucose uptake in WAT in AS160 knockout rats [54], suggesting the presence of another mechanism for glucose uptake. Similarly, canonical downstream targets of LepR [7,8], STAT3 and STAT5, were not activated by EREG within 15 min (Figure 4C). The kinetics of STAT3 and STAT5 phosphorylation in response to different EREG concentrations (Appendix A) suggest that these pathways are unlikely mediators of EREG-dependent glucose uptake. We found a rapid, but transient, increase in ERK phosphorylation in response to different concentrations of EREG in 3T3-L1 cells (Figure 4C). Phosphorylation of ERK correlated with EREG doses 15 min after stimulation (Figure 4D). Since both EGFR and LepR can phosphorylate ERK, we investigated the participation of these receptors in EREG responses. We compared the effects of EGF, a canonic ligand for EGFR, and EREG in the presence of the EGFR inhibitor AG1478 and anti-LepR antibody (Figure 4E). Both EREG and EGF induced ERK phosphorylation, though the EGFR inhibitor suppressed EGF-mediated ERK phosphorylation while failing to block EREG-mediated phosphorylation of ERK. Vice versa, anti-LepR antibodies efficiently blocked EREG-mediated phosphorylation of ERK, whereas EGF-mediated phosphorylation of ERK was not influenced by LepR inhibition (Figure 4F). In these experiments, EGF and EREG utilized distinct receptors: EGF acted via EGFR, whereas EREG activated LepR to mediate early activation of ERK.

EREG-dependent stimulation of glucose uptake also occurred in other cell types generated from peripheral tissues. EREG induced a 400% increase in FD-glucose uptake, which is significantly higher than the uptake induced by insulin under the same experimental conditions (140%) in C2C12 mouse myoblasts (Appendix A). Glucose uptake in these cells depends on EREG concentrations (Appendix A). These results indicated that EREG activated the peripheral LepR/ERK/PI3K/GLUT4 pathway to mediate glucose uptake.

### 3.5. EREG Interact with LepR In Vitro

We compared binding kinetics between recombinant LepR and leptin (Figure 5A) and LepR and EREG (Figure 5B) in vitro using quartz crystal microbalance experiments (QCM). Gold sensor changes in the frequency (ΔF) are directly proportional to the added mass and thickness in response to a generated acoustic wave. The changes in the dissipation energy (ΔD) contain information about the viscoelastic properties of the film [55]. 

The binding of leptin further increased the dissipation and decreased the frequencies seen with LepR, indicating the formation of the film on the gold surface. It was observed that after binding of LepR to gold, the final thickness value was 23 nm. When EREG was added, the thickness decreased to 20 nm indicating a mass loss in the surface due to molecular interaction and consecutively washing off molecules of LepR from the gold sensor. Thus, EREG can interact with LepR in vitro. Next, we performed a modeling analysis to elucidate how EREG can bind to LepR. 

### 3.6. EREG Evolutionary Evolved as an Alternative Ligand for LepR

A deep evolutionary analysis of the Ereg gene in 175 vertebrate species revealed highly conserved amino acids that contribute to three disulfide bonds, several polar basic residues, and a few hydrophobic amino acids (Figure 6A). 

Assessment of the conservation alongside known structures with EGFR demonstrated H78, R102, and H105 to be critical in protein interactions (Figure 6B). Docking models of EREG to LepR [5] show clustering of a likely conformation (Figure 6C) that is independent of the leptin binding sites on multimeric complex formation (Figure 6D), and dependent on H78, R102, and H105 amino acids (Figure 6E). Thus, the structural compatibility between EREG and LepR, combined with in vitro and in vivo studies, suggested that, in addition to its known role as an EGFR ligand, EREG functions as an atypical LepR ligand to regulate glucose uptake (Figure 6F).

## 4. Discussion

Our study provides structural, functional, and evolutionary evidence for LepR binding with EREG. The binding of EREG to LepR was demonstrated by immunoprecipitation of this complex in the peripheral adipose tissue of *Lep^ob^* mice under physiological conditions, upon stimulation with EREG, as well as by the interaction of EREG and LepR on a gold sensor in QSM. In the absence of leptin, the canonic ligand for LepR, EREG binds with LepR and activates a signaling cascade, resulting in glucose uptake by peripheral tissues and systemic improvement in glucose metabolism. The glycemic effects of EREG occurred in both mice and humans in different types of cells, e.g., preadipocyte and muscle cells. Although EREG is a widely established ligand for EGFR [16], its glycemic effects were not disrupted by the inhibition of EGFR, other ERB receptors, or insulin/IGFR pathways in our study. Moreover, the EREG antibody appeared to eliminate the glycemic effects of another EGFR ligand, EGF. Previous reports demonstrated EREG secretion was induced by EGF/EGFR axes; however, the role of EREG response in glucose metabolism has not been considered [16]. Notably, LepR deficiency abolished the glycemic action of EREG in multiple systems in vitro and in vivo. Moreover, we showed that LepR can interact with EREG in vitro. Although our study cannot rule out the possibility that the multimeric LepR complex still includes EGFR, our data support the key role of LepR in the glycemic action of EREG whereas EGFR appears to be dispensable for glycemic function in fibroblasts representing peripheral tissue responses. Computational structure analysis revealed that alternative EREG binding sites were conserved throughout the evolution of LepR in different species, in addition to the typical leptin binding sites. The role of the pair of ligands, leptin and EREG, in LepR regulation, appears to provide an evolutionary advantage in securing glucose uptake under leptin-deficient conditions. 

The differences in EREG and leptin binding to LepR likely contribute to different functional effects of these ligands. Leptin binds to LepR via CHR2 domains, leading to the obligatory activation of IGD and FN III domains [6]. The binding results in conformational changes in the long intracellular domains of the LepRb isoform required for hypothalamic STAT signaling [8,59]. A wide range of biological processes, including glucose uptake, energy expenditure, food intake, reproduction, and immune responses depend on leptin/LepRb signaling [1,60]. EREG treatment in *Lep^ob^* mice on regular or high-fat diets had a principal effect on glucose uptake without changing weight or energy expenditure. This lack of classical hypothalamic action of long LepR isoform on body weight and energy expenditure in the presence of EREG suggests that atypical EREG binding is not sufficient to phosphorylate tyrosine and activate the STAT3 pathway, which is required for this action in the brain [61]. In agreement, in our cell culture studies, EREG did not activate the STAT3 pathway. Low levels of *Ereg* expression have been reported in the hypothalamus and recent studies have shown the association of EREG with pain and allodynia [62]. However, the functional presence of EREG in the brain has not been reported and it is unknown if EREG can pass the blood–brain barrier. A recent study [63] suggested that sequential stimulation of the LepR–EGFR complex by leptin and EGF in tanycytes underlined leptin transcytosis in the brain; however, a transcytosis of EGFR ligands has not been investigated. It is possible that intracerebroventricular administration of EREG could affect food intake and metabolism; however, these studies were beyond the scope of our study, focusing on glycemic effects of EREG in fibroblasts of WAT and its systemic responses.

The importance of peripheral LepR signaling has recently been highlighted in a study comparing LepR deficiency in neural tissues with an overall *Lep^ob^* phenotype [64]. LepR deficiency in neural tissues only partially replicates glycemic and other metabolic abnormalities observed in *Lep^ob^* male mice, whereas regulation of appetite and body composition depends entirely on the leptin action in neurons via the long isoform of LepR (LepRb) [64]. These differences could be attributed to the presence of shorter, peripheral isoforms of LepR (i.e., LepRa, LepRc, LepRd, LepRf), which are ubiquitously expressed in peripheral tissues and are capable of increasing ERK2 phosphorylation [8]. In our experiments in cells representing peripheral tissues, EREG binding to LepR was associated with distinct signaling; EREG transiently activated ERK and required PI3K activation for efficient glucose uptake. In these cells, EREG did not induce STAT signaling mediated by the long form of LepRb [8,59]. Although the investigation of EREG binding to different LepR isoforms was beyond the scope of this study, it is likely that EREG binding to alternative sites of LepR could only mediate glycemic effects and be uncoupled from other functions regulated by the leptin/LepRb complex in the hypothalamus. The uncoupling of immune signaling from metabolic responses has been previously reported in mice with a mutant IGD domain [14] interacting with EGFR [15]. To maintain focus on glycemic effects, inflammatory responses were not studied here. We demonstrated a new functional dimension of LepR activated by EREG, mainly the regulation of glucose uptake in cells of peripheral tissues. 

We also assessed the efficacy of EREG in glucose uptake using tissues of severely obese individuals (BMI = 50 ± 8.4). EREG significantly induced glucose uptake in a dose-dependent manner, at lower concentrations than leptin or insulin. Nonetheless, these studies did not provide a conclusive answer to the relative contribution of EREG to LepR signaling under leptin-sensitive or leptin-resistant conditions, which must be addressed in future studies. The identification of glucose uptake mediators overcoming leptin and insulin resistance could provide new strategies for the treatment of the major form of type 2 diabetes mellitus. 

The proposed duality of EREG functioning as a ligand for either EGFR or LepR may shed light on EREG’s functions in tumorigenesis [43,44]. EREG forms less stable EGFR dimers than EGF [16], leading to leptin secretion [17], and rendering EREG less mitotically active in comparison with the other members of the EGF family [27]. These unique ‘loose’ binding characteristics of EREG with EGFR might facilitate its binding with LepR, increasing glucose influx in normal and/or cancer cells.

## 5. Conclusions

Most signaling pathways, including those associated with the insulin receptor [65] and EGFR [16], function with multiple ligands. The critical role of LepR in glucose metabolism may also depend on the joint action of leptin and EREG in nervous and peripheral tissues, which optimizes glucose utilization when challenged physiologically, pathologically, and/or environmentally.

## Figures and Tables

**Figure 1 cells-11-00425-f001:**
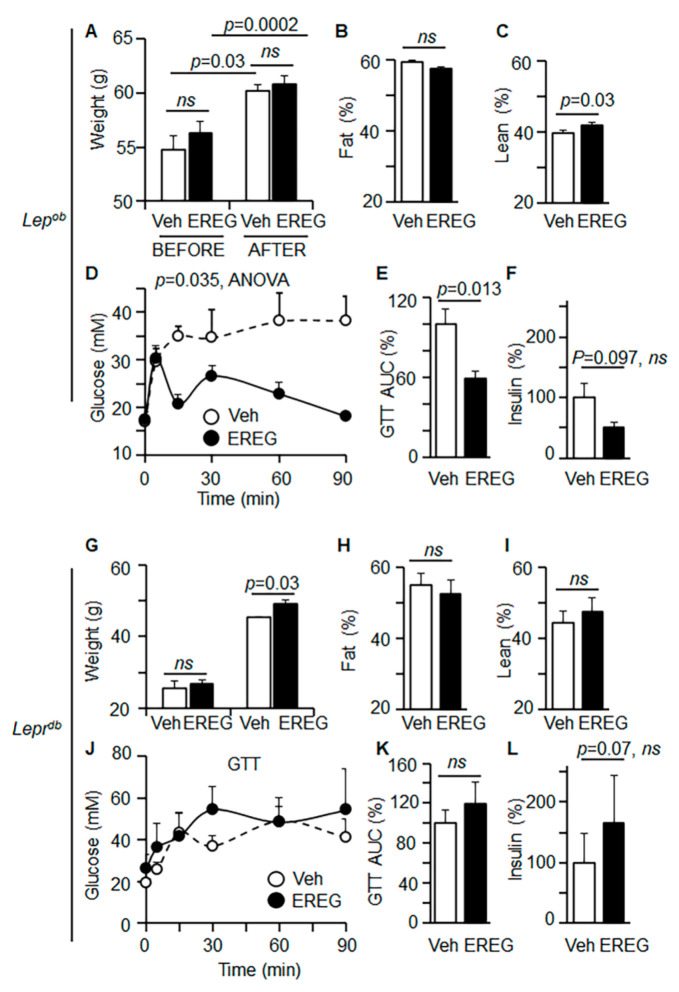
EREG improved glucose tolerance in the absence of leptin in *Lep^ob^* mice and exhibited no effect in LepR-deficient *Lepr^db^* mice. (**A**) Body weight of *Lep^ob^* male mice in groups before and after treatment with Veh (PBS, white bar) or EREG (50 ng/g body weight (BW), black bar) for 26 days. Mice were on regular chow diet. Unpaired *t*-test, *n* = 7/group. ns, not significant. (**B**,**C**) Fat (**B**) and lean body (**C**) composition in same groups of mice at the end of the study was measured by Echo-MRI. Fat and lean mass are shown as % of the total weight (100%). (**D**,**E**) Glucose tolerance test (GTT) was performed in fasted *Lep^ob^* mice treated with PBS (Veh, open circles) or EREG (closed circles) (*n* = 7 per group). GTT kinetics (**D**) and area under the curve (AUC) (**E**) are shown. Statistical significance was examined by ANOVA (**D**) and Student’s *t*-test (**E**). (**F**) Insulin levels in plasma in both mouse groups were measured by ELISA. Unpaired student’s *t*-test. (**G**) Weight before and after treatment of *Lepr^db^* male mice with Veh (PBS, white bar) or EREG (50 ng/g body weight (BW), black bar) for 4 weeks (*n* = 6 per treatment). Mice were on regular chow. Unpaired Student’s *t*-test, *n* = 6/group. (**H**,**I**) Fat (**H**) and lean body (**I**) composition (% of total weight) in the same groups of mice at the end of the study were measured by Echo-MRI. (**J**,**K**) GTT kinetics (**J**) and AUC (**K**) were obtained from *Lepr^db^* mice treated with PBS (Veh, open circles) or EREG (closed circles). ANOVA (**J**) and Student’s *t*-test (**K**). (**L**) Insulin levels in plasma in both mouse groups were measured by ELISA. Unpaired student’s *t*-test.

**Figure 2 cells-11-00425-f002:**
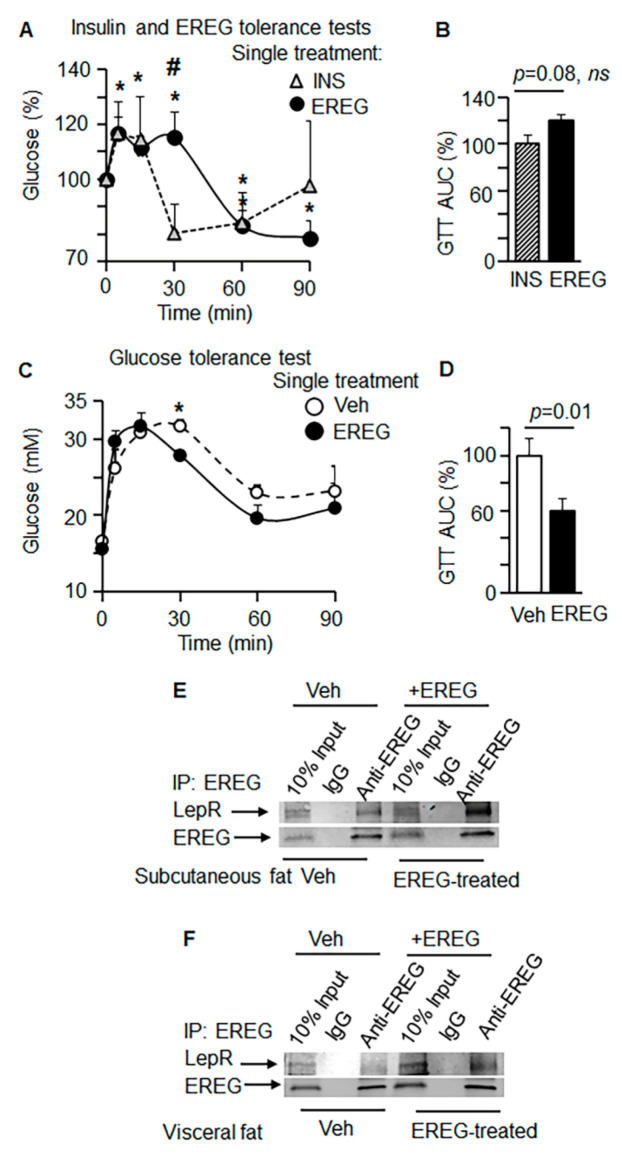
EREG regulated glucose uptake via binding with LepR in *Lep^ob^* mice. (**A**) EREG and insulin tolerance test in *Lep^ob^* mice (*n* = 5 per group) treated with a single intraperitoneal injection of insulin (0.012 IU/g BW, triangle dashed line) or EREG (80 ng/g BW, closed circles. Asterisks, significant (* *p* < 0.05) compared to glucose levels before EREG treatment. # Hashtag, significant difference in glucose levels 30 min after treatment with EREG or insulin. Unpaired Student’s *t*-test. (**B**) Area under the curve (AUC) quantification of insulin (hatched bar) and EREG (black bar) tolerance tests. Unpaired Student’s *t*-test, *ns*. (**C**) GTT kinetics were measured in *Lep^ob^* mice (*n* = 5 per treatment) treated with a single injection of PBS (Veh, open circles) or EREG (closed circles). Student’s *t*-test. * *p* < 0.05 from comparison between control and EREG treated mice at each time point. (**D**) Area under the curve (AUC) quantification of insulin (hatched bar) and EREG (black bar) tolerance tests. Unpaired Student’s *t*-test. (**E**,**F**). Immunoprecipitation of LepR was performed with anti-EREG antibody using homogenates from subcutaneous fat (**C**) and visceral fat (**D**). Fat tissue was isolated from non-treated *Lep^ob^* (Veh) as well as *Lep^ob^* mice 15 min after injection of EREG (50 ng/mL).

**Figure 3 cells-11-00425-f003:**
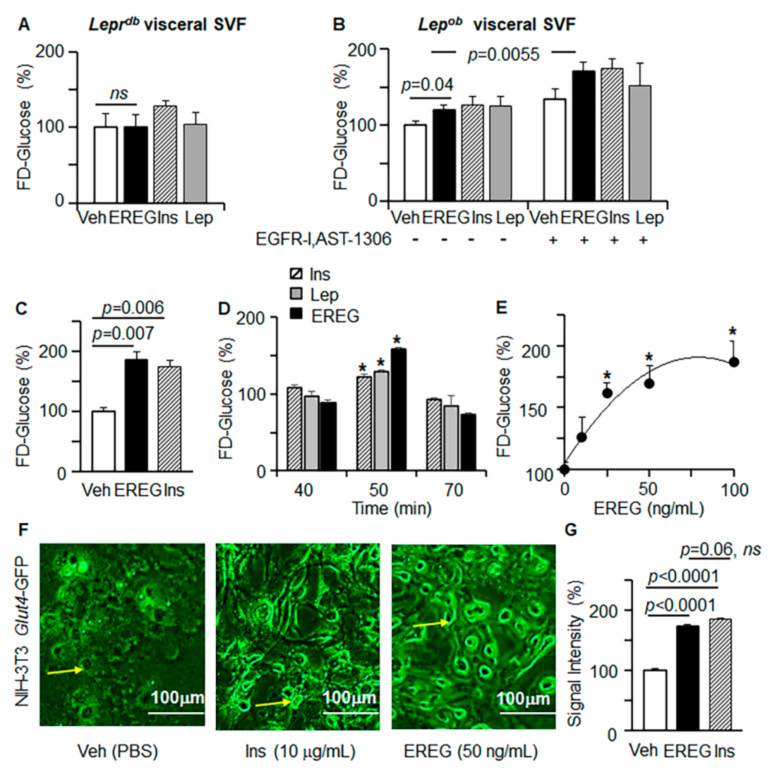
EREG-stimulated glucose uptake was dependent on LepR but independent of EGFR. (**A**,**B**) Fluorescently-labelled (FD) glucose uptake was measured in stromal vascular fraction (SVF) cells isolated from visceral tissues of *Lepr^db^* (**A**) or *Lep^ob^* mice (**B**). Cells were treated with either Veh (PBS), mouse EREG (50 ng/mL), human insulin (Ins, 10 µg/mL), or mouse leptin (Lep, 200 ng/mL) for 80 min. For inhibition experiment, *Lep^ob^* SVF cells were pre-treated with EGFR inhibitor (EGFR-I, AST-1306, 10 µM) or vehicle (Veh, DMSO) for 40 min. Data are shown as a percentage of Veh-treated control (100%, *n* = 8 per treatment). Unpaired Student’s *t*-test. (**C**–**E**) FD-glucose uptake was measured in mouse 3T3-L3 preadipocytes. (**C**) Preadipocytes were treated with vehicle, human insulin (Ins, 10 µg/mL), and mouse EREG (50 ng/mL) for 30 min (mean ± SEM, *n* = 6, *t*-test). (**D**) Time-dependent uptake of FD-glucose in 3T3-L1 preadipocytes stimulated with human insulin (Ins, 10 µg/mL), mouse leptin (Lep, 200 ng/mL), and mouse EREG (50 ng/mL). Data are shown (mean ± SEM, *n* = 8, *t*-test) as % of glucose uptake compared to control cells at the same time point (Veh, 100%). (**E**) Concentration-dependent increase in FD-glucose uptake by 3T3-L1 preadipocytes stimulated with different concentrations of mouse EREG. Data are shown as a percentage of Veh-treated control (100%, *n* = 6 per concentration). * *p* < 0.05, significant differences compared to the vehicle group, one-way ANOVA). (**F**) NIH-3T3 preadipocytes were transiently transfected with pB-*Glut4*-7myc-GFP and stimulated with vehicle, Ins (10 µg/mL), EREG (50 ng/mL) for 60 min. Data show representative fluorescent images of GFP-labeled GLUT4 selected from three independent experiments. 10× magnification. Yellow arrow shows GFP-labeled GLUT4 that was translocated to the cellular membrane. (**G**) Quantification of GFP was performed in adipocytes of similar size (*n* = 10) in each group.

**Figure 4 cells-11-00425-f004:**
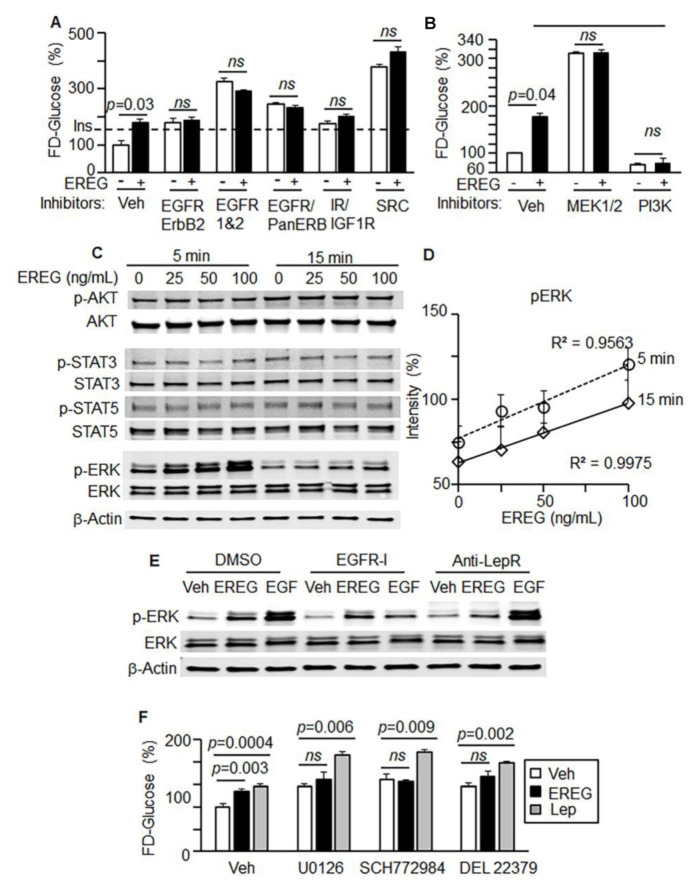
EREG mediates glucose uptake via PI3K with transient activation of ERK. (**A**) FD-glucose uptake in 3T3-L3 preadipocytes treated with or without EREG (50 ng/mL) and in the presence of inhibitors for EGFR-I (AG1478, 10 µM), EGFR and ErbB2 (AST-1306 or CI-1033 10 µM), dual IR/IGF-1R inhibitor (BMS 536924, 1 µM), and SRC-I, AZM475271, 1 µM) for 30 min. Cells were starved for 90 min before stimulation. Dashed line shows glucose uptake in the presence of insulin (Ins, 10 µg/mL). (**B**) FD-glucose uptake was measured in mouse 3T3-L1 preadipocytes with or without EREG (50 ng/mL) and inhibitors of MEK1/2 and PI3K (MEK1/2-I, U0126 10 μM, and PI3K-I, wortmannin 200 nM). Data (mean ± SD, *n* = 6) are shown as a percentage of control (Veh 100%). Unpaired Student’s *t*-test. (**C**) 3T3-L1 preadipocytes were stimulated with EREG at different concentrations (0–100 ng/mL) for 5 or 15 min. The total and phosphorylated levels of AKT, STAT3, STAT5, and ERK were measured by Western blot in duplicates. Data are shown in a representative Western blot. (**D**) The kinetics of pERK expression was quantified based on the Western blots. pAKT, p-STAT3, and p-STAT5 analysis are described in Appendix A. Pearson correlation analysis. (**E**) 3T3-L1 preadipocytes were stimulated with or without EREG or EGF (50 ng/mL, each) for 30 min in the presence and absence of EGFR inhibitor AST1306 (100 nM), and antibody against mouse LepR (Invitrogen, PA1-053, 10 μg/mL). For inhibition, cells were pre-treated 30 min before EREG and EGF stimulation. (**F**) FD-glucose uptake was measured in mouse 3T3-L3 preadipocytes pre-treated with either Veh (DMSO) or ERK inhibitors (U0126, SCH772984, or DEL 22379, each 10 µM in DMSO) for 40 min. Then, cells were treated with either Veh (PBS), mouse EREG (50 ng/mL), or mouse leptin (Lep, 200 ng/mL) for 80 min. Data are shown as a percentage of Veh-treated control (100%, *n* = 7 per group). Unpaired Student’s *t*-test. *ns*, not significant (*p* > 0.05).

**Figure 5 cells-11-00425-f005:**
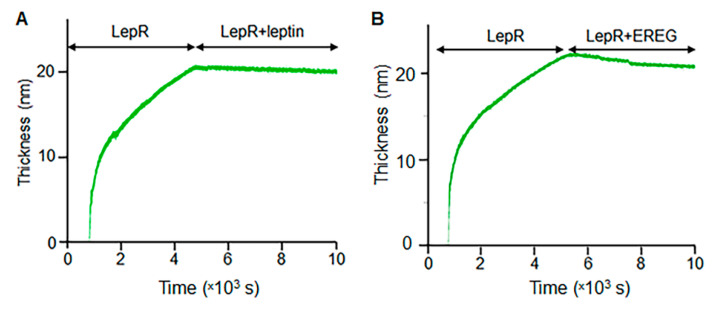
Kinetics of the changes in LepR film thickness in the presence of leptin (**A**) or EREG (**B**). Film thickness was measured using QCM and quantified based on the binding kinetics to a gold sensor.

**Figure 6 cells-11-00425-f006:**
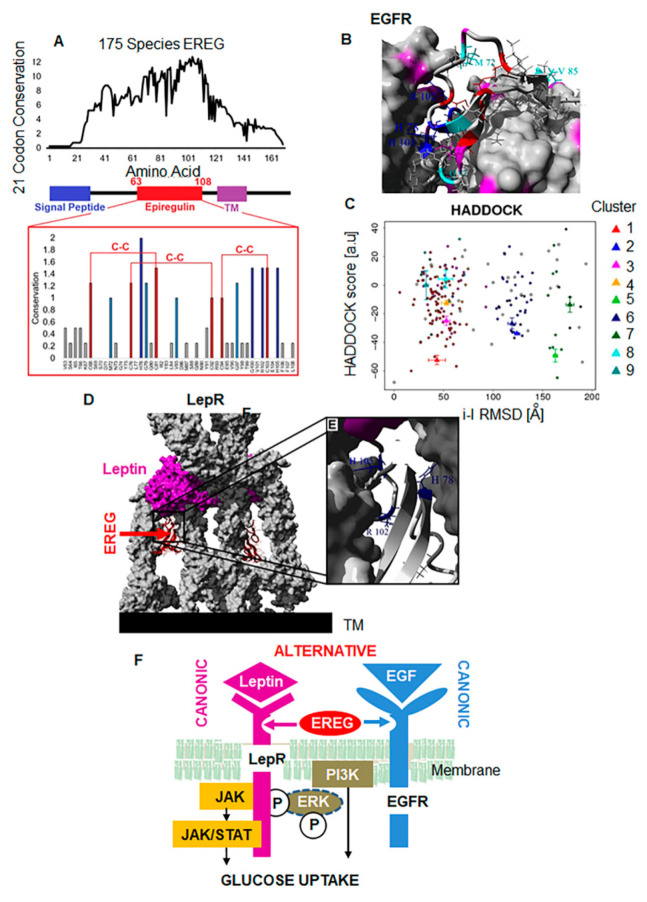
Evolutionary analysis of EREG binding to LepR. (**A**–**E**) EREG docking to LepR. Evolutionary analysis of 175 open The dependence of EREG-mediated glucose uptake on the ERK phosphorylation cascade was examined using (1) a specific inhibitor of ERK1/2 SCH772984 [56], (2) an inhibitor of ERK dimerization DEL-22379 [57], and (3) a selective inhibitor of MEK1 and MEK2 U0126 [58]. All inhibitors increased basal glucose uptake, which was further increased by leptin (Figure 4F). The inhibition of ERK1/2 and MEK1/2 as well as ERK dimerization prevented stimulatory effect of EREG on FD-glucose uptake but did not decrease it beyond the levels seen in the control cells. Although transient ERK phosphorylation occurred in response to EREG stimulation, this pathway was dispensable for glucose uptake and dependent on PI3K and may be other pathways (Figure 4B and Appendix A). (**F**) Hypothetic mechanism suggesting EREG as an alternative ligand for both EGFR and LepR. The canonic leptin/LepR response can induce JAK/STAT3 signaling and required the long form of LepR. The alternative binding of EREG to LepR can induce ERK and PI3K activation increasing GLUT4 translocation and glucose uptake, but not the other canonic effects of leptin, including the regulation of appetite and energy expenditure.

## Data Availability

Not applicable.

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
