# Peer review of "Epiregulin as an Alternative Ligand for Leptin Receptor Alleviates Glucose Intolerance without Change in Obesity"

_cells, 2022, doi:10.3390/cells11030425_

Round 1

Reviewer 1 Report

This is an interesting manuscript where the authors analyze the role of epiregulin (EREG) on glucose uptake in peripheral tissues and its possible interaction with LepR. I have several comments:

  1. The introduction lacks a molecular description of EREG (how many amino acids does it have?, what is known about its structure?, what tissue secretes it?)
  2. In the experiments summarized in Fig. 1, various effects of EREG are analyzed in Lep (ob) mice (leptin deficient mice) and in Lepr (ob) (leptin receptor deficient mice). The most important effect is an increase in glucose uptake only in Lep (ob) mice. In this set of experiments the following doubts arise: a) Why was that EREG concentration (50ng / g) used? b) As the effect of EREG is attributed to an interaction with Lepr (no effects are observed in Lepr-ob), in my opinion the positive control with leptin is lacking to rescue the effects in Lep (ob) and compare with EREG effects.
  3. In the discussion it says: "The direct binding of EREG to LepR was demonstrated by immunoprecipitation" ... In general, immunoprecipitation experiments performed on a tissue lysate as in this case, can only indicate that both proteins are part of a complex protein but not a direct interaction.
  4. In the discussion it says: “Taken together, these data suggest a limited interaction of EREG with the hypothalamic isoform of long LepRb”… Although this is only a suggestion, the research carried out does not show experiments with hypothalamic tissues or neuronal cultures. Another question is whether the putative EREG binding site in Lepr is conserved in the long isoform of LepRb? If so, why can't EREG bind at the brain level?
  5. With reference to Fig. 4A, these results are discussed very briefly in the manuscript. It is observed that the different inhibitors (especially EGFR / 1 & 2 and SRC), significantly stimulate glucose uptake with or without EREG. The authors should explain these results in greater detail. The inhibitors used, such as specificity, also need to be described in more detail to rule out effects on other receptors.

Author Response

We would like to thank this reviewer for the important comments improving the quality of our manuscript. Please use the pdf file to assess Figures.

Reviewer 1: This is an interesting manuscript where the authors analyze the role of epiregulin (EREG) on glucose uptake in peripheral tissues and its possible interaction with LepR. I have several comments:

Reviewer 1: 1. The introduction lacks a molecular description of EREG (how many amino acids does it have?, what is known about its structure?, what tissue secretes it?)

Response: We added this information to the Introduction (Lines 81-94, four new references). This new information strengthened our model, since The Human protein Atlas demonstrates that the major expression site of EREG are tissues of mesenchymal origin, such as bone marrow and lymphoid issues (the gray bar, Reviewer Fig.1). Our studies in non-differentiated 3T3-L1 fibroblasts and stromal vascular cells are relevant because they have mesenchymal origin.

Figure 1. Comparison of EREG expression in different tissues. Gray bar demonstrates the highest expression levels in mesenchymal bone marrow and lymphoid tissues.

Reviewer 1: 2. In the experiments summarized in Fig. 1, various effects of EREG are analyzed in Lep (ob) mice (leptin deficient mice) and in Lepr (ob) (leptin receptor deficient mice). The most important effect is an increase in glucose uptake only in Lep (ob) mice. In this set of experiments the following doubts arise: a) Why was that EREG concentration (50ng / g) used? b) As the effect of EREG is attributed to an interaction with Lepr (no effects are observed in Lepr-ob), in my opinion the positive control with leptin is lacking to rescue the effects in Lep (ob) and compare with EREG effects.

Response: We used 50ng/g BW EREG based on our dose-dependence studies in cell cultures (MS Fig. 3E). We did not use leptin as a positive control in the current MS, because ob/ob mouse model is one of the most studied models of insulin resistance (the search with ‘GTT ob/ob insulin resistance’ yields >500,000 PMID articles) and the role of leptin in the compensating for glucose tolerance is well established. Our previous work demonstrated that leptin improves glucose tolerance in the ob/ob mice (Reviewer Fig 2, ref https://doi.org/10.1371/journal.pone.0153198). However, in response to the reviewer’s concern (see also answers to the next question), we compared the binding kinetics of leptin vs. EREG to the recombinant LepR in vitro (New Fig.5 in the revised MS).

Reviewer 1: 3. In the discussion it says: "The direct binding of EREG to LepR was demonstrated by immunoprecipitation" ... In general, immunoprecipitation experiments performed on a tissue lysate as in this case, can only indicate that both proteins are part of a complex protein but not a direct interaction.

Response: To strengthen our interpretation, we performed a new experiment (new Fig.5).  The interaction of LepR bound to gold sensors was measured with recombinant leptin or EREG as a change in frequencies and dissipation of energy, that was allowed to calculate the film thickness. Our data revealed that EREG retouched the LepR from the gold sensor, suggesting its direct interaction with LepR in vitro. Regardless, we agree with this reviewer that our data required more cautious interpretation. In the revised MS (Lines 535-538), we re-phrased this sentence by removing ‘direct’ and adding the in vitro findings described in Fig. 5.

Reviewer 1: 4. In the discussion it says: “Taken together, these data suggest a limited interaction of EREG with the hypothalamic isoform of long LepRb”… Although this is only a suggestion, the research carried out does not show experiments with hypothalamic tissues or neuronal cultures. Another question is whether the putative EREG binding site in Lepr is conserved in the long isoform of LepRb? If so, why can't EREG bind at the brain level?

Response: We would like to thank this Reviewer for this important comment.  We deleted this misleading statement. EREG binds to extracellular region oh LepR (Fig. 6, revised MS) that is identical in long and short forms of LepR. EREG can potentially bind to cognate receptors in the hypothalamus since it could be expressed in this brain region, according to Human Protein Atlas. However, in our studies EREG did not activate STAT3 pathways, which is required for hypothalamic action of leptin on food intake and energy expenditure (ref. 61). EREG binds to the  alternative site on cytosolic domain of LepR, which induce another pathway. We discuss this in the Lines 577-596:

This lack of classical hypothalamic action of long LepR isoform on body weight and energy expenditure in the presence of EREG suggests that atypical EREG binding is not sufficient to activate tyrosine phosphorylation and activation of STAT3 pathway, which is required for this action in the brain [61]. In agreement, in our cell culture studies, EREG did not activate STAT3 pathway. Low levels of Ereg expression has been reported in the hypothalamus and recent studies have shown association of EREG with pain and allodynia [62]. However, the functional presence of EREG in the brain has not been reported and it is unknown if EREG can pass blood-brain barrier. Recent study [63], suggested that sequential stimulation of LepR–EGFR complex by leptin and EGF in tanycytes underlined leptin transcytosis in the brain; however, a transcytosis of EGFR ligands has not been investigated. It is possible that intracerebroventricular administration EREG could affect food intake and metabolism; however, these studies were beyond the scope of our study, focusing on glycemic effects of EREG in fibroblasts of WAT.

The importance of the peripheral LepR signaling has recently been highlighted in a study comparing LepR deficiency in neural tissues with an overall Lepob phenotype [64]. LepR deficiency in neural tissues only partially replicates glycemic and other metabolic abnormalities observed in Lepob male mice, whereas regulation of appetite and body composition depends entirely on the leptin action in neurons via long isoform of LepR (LepRb)  [64].

Reviewer 1: 5. With reference to Fig. 4A, these results are discussed very briefly in the manuscript. It is observed that the different inhibitors (especially EGFR / 1 & 2 and SRC), significantly stimulate glucose uptake with or without EREG. The authors should explain these results in greater detail. The inhibitors used, such as specificity, also need to be described in more detail to rule out effects on other receptors.

Response: This information is now added in Lines 415-457 together with the new references.

Reviewer 2 Report

The manuscript "Epiregulin as an alternative ligand for leptin receptor alleviates glucose intolerance without change in obesity" by song et al. is interesting and the hypothesis of the activation of leptin receptor by Epiregulin in plaussible. Nevertheless, there are a few flaws in the design of the experiments and interpretation that prevent the manuscript ftom being accepted in the present form.

  1. One of the conclusions of the study is that Epiregulin increases glucose uptake acting through leptin receptors without modifying food intake and wheight. However, the administration of Epiregulin was intraperitoneal and the transport of Epiregulin across the blood brain barrier is not known. Therefore, intracerebroventricular administration could affect food intake and metabolism.
  2.  The mechanistic explanation of Epiregulin effects on glucose is based on animal models without leptin receptors and in vitro studies with immunoprecipitation. Nevertheless it is well known the presence of EGFR/Leptin receptor heterodimers, and co-immunoprecipitation of Epiregulin with leptin receptor may be mediated by the binding to EGFR. In this line, Epiregulin could interact with EGFR (EGFR-1 or EGFR´2) even though the effect on glucose may require the presence of leptin receptors. 
  3. Even though the study of the leptin receptor isoform activated by leptin was not supposed to be an aim of this study, at least a direct effect of Epiregulin activation of leptin receptor should be provided, for instance, the analysis the Tyr-phosphorylation of leptin receptor upon Epiregulin stimulation should be provided.

Author Response

Reviewer 2

Reviewer 2. The manuscript "Epiregulin as an alternative ligand for leptin receptor alleviates glucose intolerance without change in obesity" by song et al. is interesting and the hypothesis of the activation of leptin receptor by Epiregulin in plaussible. Nevertheless, there are a few flaws in the design of the experiments and interpretation that prevent the manuscript ftom being accepted in the present form.

We would like to thank this reviewer for the important comments improving the quality of our manuscript.

Reviewer 2: 1. One of the conclusions of the study is that Epiregulin increases glucose uptake acting through leptin receptors without modifying food intake and wheight. However, the administration of Epiregulin was intraperitoneal and the transport of Epiregulin across the blood brain barrier is not known. Therefore, intracerebroventricular administration could affect food intake and metabolism.

Response: Thank you for pinpointing this important aspect of the regulation. We revised discussion in Lines 577-589:

This lack of classical hypothalamic action of long LepR isoform on body weight and energy expenditure in the presence of EREG suggests that atypical EREG binding is not sufficient to activate tyrosine phosphorylation and activation of STAT3 pathway, which is required for this action in the brain [61]. In agreement, in our cell culture studies, EREG did not activate STAT3 pathway. Low levels of Ereg expression has been reported in the hypothalamus and recent studies have shown association of EREG with pain and allodynia [62]. However, the functional presence of EREG in the brain has not been reported and it is unknown if EREG can pass blood-brain barrier. Recent study [63], suggested that sequential stimulation of LepR–EGFR complex by leptin and EGF in tanycytes underlined leptin transcytosis in the brain; however, a transcytosis of EGFR ligands has not been investigated. It is possible that intracerebroventricular administration EREG could affect food intake and metabolism; however, these studies were beyond the scope of our study, focusing on glycemic effects of EREG in fibroblasts of WAT.

Reviewer 2: 2. The mechanistic explanation of Epiregulin effects on glucose is based on animal models without leptin receptors and in vitro studies with immunoprecipitation. Nevertheless it is well known the presence of EGFR/Leptin receptor heterodimers, and co-immunoprecipitation of Epiregulin with leptin receptor may be mediated by the binding to EGFR. In this line, Epiregulin could interact with EGFR (EGFR-1 or EGFR´2) even though the effect on glucose may require the presence of leptin receptors. 

Response: The ability of LepR to form multiprotein complex is well known and it is possible that EGFR is still a part of multimeric complex regulating glucose. Since crystal structure of multi-protein complex has not been resolved, we could not rule out participation of EGFR in the regulation of LepR in response to EREG in vivo. We modified discussion in Lines 573-584:

Moreover, we showed that LepR can interact with EREG in vitro. Although our study cannot rule out the possibility that multimeric LepR complex still includes EGFR, our data support the key role of LepR in the glycemic action of EREG whereas EGFR appears to be dispensable for glycemic function in fibroblasts representing peripheral tissue responses.

Reviewer 2: 3. Even though the study of the leptin receptor isoform activated by leptin was not supposed to be an aim of this study, at least a direct effect of Epiregulin activation of leptin receptor should be provided, for instance, the analysis the Tyr-phosphorylation of leptin receptor upon Epiregulin stimulation should be provided.

Response: We provided the new Fig. 5 demonstrating interaction between LepR and EREG, which was compared to LepR and leptin interaction (lines 503-521).  This experiment has shown direct interaction between EREG and LepR requested by the reviewer.

The known tyrosine phosphorylation occurred on the long LepR isoform in the brain (reviewed in https://doi.org/10.1038/ijo.2008.232) that is not available in short LepR isoforms expressed 3T3-L1 cells. Therefore, we studied the canonic downstream effect, namely phosphorylation of STAT3. However, EREG stimulation of 3T3-L1 did not change STAT3 phosphorylation (Fig. 4C), suggesting a unique mechanism of EREG in 3T3-L1 cells.

Round 2

Reviewer 1 Report

The revised version of the manuscript improved the aspects mentioned above. The authors included new experiments that support the results of interaction between EREG and the leptin receptor, they also included details of the use and effect of the inhibitors used in Fig. 4., among other aspects. Therefore, I support the publication of the revised manuscript, and I congratulate the authors for the improvements and experiments made in a short time.

Reviewer 2 Report

The authors have followed my recommendations and the manuscript may be published in the present form